# Revisiting Water Supply Rule Curves with Hedging Theory for Climate Change Adaptation

**Wenhua Wan, Jianshi Zhao *** and **Jiabiao Wang**

State Key Laboratory of Hydro-Science and Engineering, Department of Hydraulic Engineering,
Tsinghua University, Beijing 100084, China; meviolet@126.com (W.W.); waterwhu@foxmail.com (J.W.)
***** Correspondence: zhaojianshi@tsinghua.edu.cn

**Abstract:** Conventional reservoir operation rule curves are based on the assumption of hydrological stationarity. The aggravating non-stationarity under the changing environment rocked this foundation. The hedging theory is one of the options for adaptive operation based on hydrological forecasts, which can provide a practical tool for optimal reservoir operation under a changing environment. However, the connections between hedging theory and rule curves are not clear. This paper establishes the linkage of rule curves and hedging theory by analyzing three fundamental problems surrounding the design of conventional rule curves, namely the law and design of water supply rule curves, the determination of flood control storage, and the division of refill and drawdown circle. The general interpretation of the conventional water supply rule curves with hedging theory is conducted. Both the theoretical analyses and the Danjiangkou Reservoir case study reveal that, based on the historical records, the rule curves can be interpreted as a specific expression of hedging theory. This intrinsic linkage allows us to propose a more general and scientific method of updating rule curves in the context of non-stationarity. On this basis, the rule-curve-based climate adaptation strategies are figured out using hedging theory. This research is helpful for practical adaptive operation of reservoirs in the changing environment.

**Keywords:** changing environment; conventional rule curves; hedging theory; non-stationarity; adaptation strategy

## 1. Introduction

In most countries of the world, precipitation has clear temporal distributions within a year, leading to seasonal patterns of streamflow with typical wet-dry cycles [1,2]. For instance, China is significantly affected by the monsoon climate, with summer precipitation accounting for over 50% of the total annual precipitation [3,4]. Large reservoirs, as important infrastructure in water resource system, are essential to regulate the fluctuating streamflow. They are expected to collect and store water that is in excess of immediate requirements and then improve water availability when there is a shortage [5]. Reservoirs designed to provide water supply for irrigated agriculture, industrial or domestic life, hydropower generation, and/or to manage floods need to be operated in a reliable and effective manner.

Practically, the reservoir operation policies are usually defined in the form of rule curves (also known as guide curves), firstly proposed by the United States Army Corps of Engineers (USACE). The rule curves are trajectories of reservoir elevations used to guide or constraint reservoir operation as a lookup-figure/function of reservoir current storage condition and the within-year time period. The rule curves are developed at the planning stage through intensive regulation of typical inflow series based on long-term inflow data, water demands, experiential judgment, and engineering standard [6]. Although easy to follow, the selection of typical inflow series, the time period division within the year,

and the flow regulation processes can be highly empirical. In most cases, the fundamental principles behind the design of rule curves are not clearly stated.

Recently, increasing shreds of evidence suggest that climate change and human interferences have significantly altered the hydrological processes, including reservoir related inflows and water demands [7–9]. This fact has posed serious challenges on the conventional rule curves, which are designed based on the assumption of hydrological stationarity. The aggravating non-stationarity under the changing environment rocked the foundation of conventional rule curves [10]. For the purpose of mitigating the impact of changing climate, a feasible option is to optimize reservoir rule curves under deterministic projected future scenarios [11–13]. Similar to the black-box models, such optimization can only produce case-by-case results rather than more generic insights or general principles.

The hedging theory is one of the options for adaptive operation based on hydrological forecasts and economic expected utility theory. It has drawn increasing research attention in recent years [14–24]. Originally, the term "hedging" in water supply reservoir represents the probability of how much water to withhold from immediately beneficial deliveries and retain in storage in an attempt to reduce a more severe shortage likely to occur in future [17]. Following this concept, the existing hedging-based operations have been applied in some real-time operation cases with special objectives, i.e., water supply [16,18], flood control [19,20], refill operation [21,22], and incorporating environmental flow [23,24]. However, these applications cannot provide direct help on how the conventional rule curves can be revised for the adaption to climate change in practice. In essence, hedging lies in balancing the marginal values of uncertain temporal/spatial benefits/costs, thus realizing the purpose of risk mitigation and optimal utilization. The conventional operation also aims at reaching a higher benefit while lower risk. Therefore, there should exist some connections between hedging theory and rule curves. But, as yet, these connections are not clear. To avoid repeated decision using different operation methods, it is of great scientific and practical significance to interpret the connections between them and provide new insights to climate change.

The aim of this paper is to develop an analytical framework to revisit conventional rule curves with hedging theory and help to design adaptive rule curves. By establishing the connections between risk-based hedging theory and empirical rule curves, one can better understand and identify the economic principles behind the design of conventional rule curves, and thus be aware of the qualitative adjustment of rule curves in the changing environment. This paper is organized as follows. Section 2 splits the rule curves into four essential aspects and presents a comprehensive interpretation of rule curves using hedging theory. Section 3 provides a real-world reservoir case in the Han River Basin in China to demonstrate the linkages. Subsequently, the adaptive strategies of rule curves under some illustrative non-stationary contexts are given in Section 4. Finally, the conclusions are made in Section 5.

## 2. Revisiting Rule Curves with Hedging Theory

Most reservoirs are built for multiple purposes. Figure 1 shows the rule curves associated with a typical multi-purpose reservoir, in which the calendar year is divided into various within-year time periods, herein two seasons and two transition periods: Non-flood season, flood season, drawdown period, and refill period. In non-flood season, water supply curves guide the release with conservation water storage ($S_{cons}$) and dead storage ($S_{dead}$) as the upper and lower limits. During normal-inflow years, all planned demands are met 100% and reservoir storage is kept above the target storage curve ($S_{target,t}$), but during drought years when storage falls below the target storage curve or even firm storage curve ($S_{firm,t}$), the release is reduced. In flood season, reservoir storage is constrained by flood limited water storage ($S_{flood}$). The maximum storage capacity ($S_{max}$) is the upper bound for flood routing. In the transition periods, the reservoir should be lowered to $S_{flood}$ at the beginning of flood season and refilled to $S_{cons}$ by the end of flood season. The curves are established at the planning stage and usually kept unchanged during the lifespan of reservoirs.

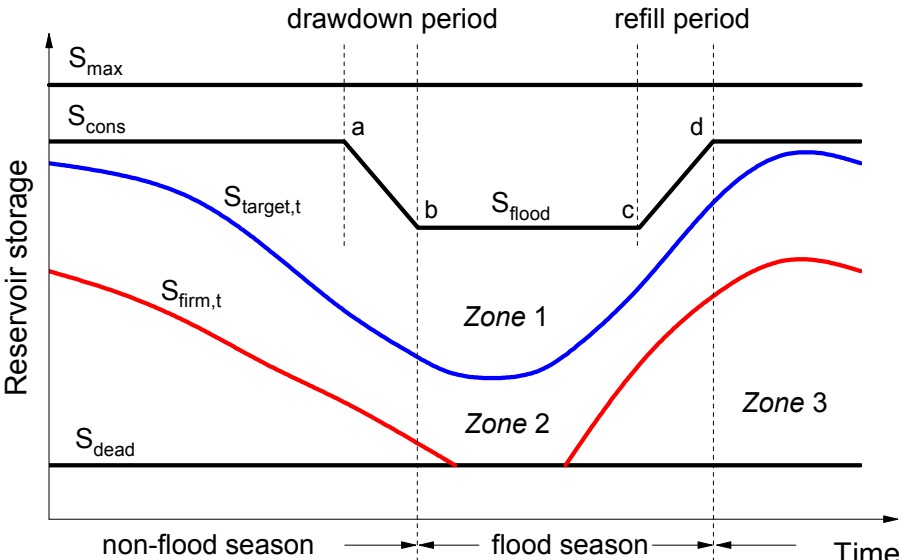

**Figure 1.** Schematic illustration of multipurpose reservoir rule curves, specifying the storage targets and the release given particular current storage and time of within-year period.

Operations in different time periods serve different purposes and, thus, require varying mathematical derivations. Based on the division of within-year operation periods, this section will disclose the estimation of conventional rule curves. Using hedging theory, the purpose of this section is to interpret the following key questions for the design of rule curves: (1) Why is water delivered by stages and why do water supply curves (i.e., $S_{target,t}$ and $S_{firm,t}$) show strong seasonality? (2) How is flood control storage (i.e., the storage between $S_{max}$ and $S_{flood}$) obtained when larger storage can be afforded? (3) When are the suitable starting points of the refill and drawdown periods?

In this paper, the focus is placed on the design of rule curves from planning perspective under given conditions including project investment, geological and topographical conditions of the dam site, reservoir size and type, etc. That is to say, the exterior boundaries of rule curves design for a reservoir/project are supposed to be known and fixed, including dead storage, conservation water storage, maximum reservoir storage, the stage–storage relationship, and historical inflow sequences. The rationality of this proposition is that the hydrological non-stationarity has little impact on these aspects.

### 2.1. Hedging Theory Formulation for Multiple-Use Water Supply Pattern

Denote by $\mathbb{S}$ the set of water demand being satisfied by reservoir release $R_t$. The engineering water supply reliability can be described by the satisfactory frequency.

$$p_{rel} = \text{Prob}(R_t \in \mathbb{S}) \tag{1}$$

Consider a reservoir supplies both domestic and agricultural water uses. When water availability is less than total demand, the strategy is always to try to guarantee the domestic water use but to cut the agricultural water supply following the reliability rates.

To explain the relative difference in reliability rates of multiple water users, it is necessary to prove the concavity of the water supply function first. Consider reservoir downstream demand involving $m$ fragmented customers, which can be commodity crops such as wheat, cotton, soybean, and maize irrigation, or industrial sectors such as papermaking, cooling, and mining, or for domestic uses such as drinking, gardening, and flushing. Denote by $b(r_i) = a_i * r_i$ $(i = 1, \ldots m,\ a_1 > a_2 > \ldots > a_m)$ the linear function benefit of each customer type. The benefit function $b(r_i)$ depends just on the amount of water being allocated to it, $r_i$. This variable should not exceed the customer water demand $d_i$; and the sum of $r_i$ cannot exceed the total reservoir release in each period, $R_t$. The water allocation planning can be described as an optimization objective to maximize the gross monetary value of water supply.

$$B(R_t) = \text{Max} \sum_{i=1}^{m} b(r_i)$$

s.t.

$$\sum_{i=1}^{m} r_i \leq R_t$$
$$r_i \leq d_i$$

(2)

Equation (2) indicates that the water allocation should follow the rule of maximizing total water supply benefits for all customers. Implementing the hill-climbing method, the optimal policy found is to allocate water from the customer with the highest marginal benefit $\partial b(r_i)/\partial r_i$, then to the next highest benefit customers, and so on, until all release has been distributed or all demands have been fulfilled. That is, the released water tends to be allocated in order of cost-effectiveness ($a_i$), leading to a rough concave and monotonic increasing economic benefit function $B(R_t)$ (i.e., decreasing marginal benefits with increasing release) [16,25]. Accordingly, the social-economic marginal benefit of domestic use is relatively higher than that of agricultural use, as well as ecological use, which explains the always larger design reliability rate for domestic water supply compared to the others. Note that the water supply social-economic benefit is not always reflected by water price.

Even operating based on rule curves, it is difficult to determine what actions are the best to take without knowing how much inflow may occur. The water supply operation is a kind of long-term decision process that should not just account for the current water demand impacts, but also compromise the ability to meet future demands. Quite a few recent works reveal that a rational hedging water supply should be a function of total water availability currently, $WA_t$, with the objective to maximize the sum of immediate benefit $B(R_t)$ and carryover storage benefits $C(S_{t+1})$ [15,16].

$$\text{Max } z = B(R_t) + C(S_{t+1})$$

s.t.

(3)

$$R_t + S_{t+1} = WA_t = S_t + I_t - E_t$$

where $S_t$ is the current-period/initial reservoir storage; $E_t$ is the evaporation and seepage losses; and $I_t$ is the anticipated inflow. In hedging theory, $I_t$ generally refers to the forecast inflow, whereas in determining rule curves, this value should be the historical observations.

As a mathematical operation of concave functions, any nonnegative weighted sum of concave functions is concave. Since the single period utility function $B(R_t)$ is concave, the economic function of storing water for future use $C(S_{t+1})$ should also be concave, as well as the total utility function z. Applying the Lagrangian condition, the optimal hedging is to equalize the marginal values of water delivery and water storage as much as possible [16] (referred to as "optimality"):

$$\frac{\partial B(R_t)}{\partial R_t} = \frac{\partial C(S_{t+1})}{\partial S_{t+1}}$$

(4)

This equation implies that hedging allows a small deficit of current target release and saves to protect against the future large deficit, thus achieving optimal overall efficiency. Considering the relationship between $R_t$ and $S_{t+1}$, and the concavities of utility functions, Zhao et al. [26] further proved that the optimal decisions, $R_t^*$ and $S_{t+1}^*$, increase in accordance with water availability $WA_t$ (referred to as "monotonicity"). Recall the optimal principle in Equation (4) and the monotonic relationship, the optimal water supply decision $R_t^*$ should be reduced when available water is small, and vice versa.

As reflected in the conventional rule curves, water supply shall be small if the current water level/availability is low. For instance, if the beginning reservoir storage is in Zone 1, all planned demand is met; but if the storage is in Zone 2 or 3, different degree of release curtailment is needed; and the lower the storage zone, the higher the reduction of water supply. This stage-wise release curtailment is consistent with the abovementioned optimal hedging rule. Similarly, the seasonality

of rule curves can also be interpreted by the optimal principle and monotonicity of hedging rule. Without loss of generality, consider two operation periods with one in flood season and the other in non-flood season (as listed in Table 1). Assume identical initial storages in the two periods, denoted by $S_t$. In general, the expected inflow in flood season is greater than that in non-flood season, denoted by $I_t + \Delta I$ and $I_t$, respectively. Then the total water availability in the two periods are $WA_t + \Delta I$ and $WA_t$. The monotonic relationship of the hedging release indicates that the optimal supply should also be larger in flood season. Consider such a release decision discriminated by the target storage curve. Then in period $t_1$, the reservoir storage should be located at somewhere higher than $S_{target,t_1}$, whereas the storage in period $t_2$ should be lower than $S_{target,t_2}$. In brief, even if the initial storage is basically the same, when the expected inflow is high, there is a lower probability to hedge (more water supply for immediate benefit) due to the higher ability of water supply and the increased burden of flood control. That is to say, in the high-flow periods, the water supply rule curves ($S_{target,t}$ and $S_{firm,t}$) shall be low, thus allowing more water to be released, and vice versa. This explains the stage-wise water supply pattern and the seasonality of rule curves.

**Table 1.** Water supply decision in different operation periods under the same initial reservoir storage.

| Operation Period | Flood Season $t_1$ | Non-Flood Season $t_2$ |
|---|---|---|
| Initial storage | $S_{t_1} = S_t$ | $S_{t_2} = S_t$ |
| Expected inflow | $I_{t_1} = I_t + \Delta I$ | $I_{t_2} = I_t$ |
| Water availability | $WA_t + \Delta I$ | $WA_t$ |
| Hedging release | $R_t^* + \Delta R$ | $R_t^*$ |
| Rule-curves release | Increased release (higher than $S_{target,t_1}$) | Firm release (lower than $S_{target,t_2}$) |

The above analyses show that water supply hedging rule can effectively interpret the fundamental characteristics of water supply rule curves. Hence, it is possible to specify rule curves using a zone-based hedging rule following Tu et al. [27,28], as given below:

$$R_t = \begin{cases} 0 & S_t < S_{dead} \\ \alpha_2 D_t & S_{dead} \le S_t < S_{firm,t} \\ \alpha_1 D_t & S_{firm,t} \le S_t < S_{target,t} \\ D_t & S_{target,t} \le S_t \le S_{up,t} \end{cases} \tag{5}$$

where $S_{up,t} = S_{cons}$ in non-flood season and $S_{up,t} = S_{flood}$ in flood season; $\alpha_1$ and $\alpha_2$ are the given rationing factors, and $0 < \alpha_2 < \alpha_1 < 1$. Note that $\alpha_1 D_t$ should be greater than or equal to reservoir firm supply according to the design requirement of rule curves. The values of $\alpha_1$ and $\alpha_2$ are determined at the planning stage but can be modified to meet updated operational purposes.

To obtain the zone-based hedging rule curves, the objective to maximize the expected utility of long-term water supply benefit is needed, which is the multi-period hedging model [18,29]:

$$\text{Max} \sum_{t=1}^{n} E[B_t(R_t)]$$

s.t.

$$S_t + I_t - E_t - R_t = S_{t+1} \tag{6}$$

where $n$ represents the total operation periods; $E[B_t(R_t)]$ is the expected utility function value of water delivery $R_t$. Similar to Equation (4), the ideal solution is that the marginal utility function of all periods to be identical, thus allocating total water supply to each period as evenly as possible:

$$\frac{\partial B(R_t)}{\partial R_t} = \frac{\partial B(R_{t+1})}{\partial R_{t+1}} \tag{7}$$

According to the hedging rule in Equation (5), the amount of water release is defined as the function of current storage. Using the long-term historical inflows as input and Equation (6) as the optimization model, the optimal hedging rule curves can be found through long-term optimization-simulation. Since the conventional rule curves are obtained based on historical sequences and the hedging theory is able to capture the characteristics of rule curves, the achieved hedging rule curves should be consistent with the conventional rule curves based on historical statistics.

### 2.2. Hedging Theory for Flood-Control Storage Allocation

Large reservoirs are one of the main structures that can mitigate the risk of downstream being damaged by floods by storing all or a portion of the floodwater in reservoir and then releasing them slowly over time. In conventional rule curves, the determination of the flood limited water level (FLWL, or flood-control storage) is based on the design flood hydrograph. The frequency of design flood is generally obtained following the engineering design standard. Regardless of the engineering standard, if reservoir functions just flood risk control, maximizing flood control storage (i.e., $S_{flood} = S_{dead}$) will be the best strategy; otherwise, if the reservoir is responsible for just economic benefits, it is better to minimize flood control storage (i.e., $S_{flood} = S_{cons}$). However, in practice, the FLWLs of most reservoirs are set between the dead level and conservation level. This indicates that an appropriate setting of FLWL needs to hedge the flood risk by diversifying economic operation.

To explain this phenomenon, we first build the linkage between flood control storage and flood risk level. Denote by $X$ the peak flow of flood events, $x$ a possible value of $X$, $F_X(x)$ the cumulative distribution function, and $f_X(x)$ the probability density function of $x$. A common method of fitting $f_X(x)$ is to plot the ranked historical annual maximum flood series on special probability paper and draw a line through these data [30]. The $(1-p)$th quantile of $X$ is $x_p$, such that $X$ has a probability $p$ being equal to or exceeding $x_p$.

$$\text{Prob}(X \geq x_p) = p = 1 - F_X(x_p) = \int_{x_p}^{+\infty} f_X(x)dx, \tag{8}$$

$$x_p = F_X^{-1}(1-p), \tag{9}$$

For any selected risk level $p_0$, the corresponding flood hydrograph can be extracted from historical floods, and sometimes homogenous frequency enlargement of typical floods is needed. According to the reservoir prescribed flood operation rule, the corresponding flood control storage $FS_{p_0}$ can be obtained by regulating this flood hydrograph. A function is defined to describe this process:

$$FS_{p_0} = Routing(x_{p_0}) \tag{10}$$

Equations (8)–(10) show the relationship between the flood control storage and flood risk level (i.e., the flood control standard). Thus, the question left is to determine how much flood control storage, if any, should be contained.

Given reservoir maximum and dead storages, we have reservoir storage capacity $K = S_{max} - S_{dead}$. This amount of storage capacity is allocated to either flood control storage $FS$ for regulating potential floods, or active storage $AS$ for long-term beneficial use.

Flood control storage is reserved for reducing potential downstream flooding damages during floods, where the more storage space a reservoir allocates, the safer downstream will be. Denote by $B_1(FS_{p_0})$ the flood prevention benefit of flood control storage which can regulate floods smaller than $x_{p_0}$. The flood prevention benefit can be valued by the expected avoidance of damages or costs, and other advantageous effects during floods due to the construction of flood control structure (i.e., reserved flood control storage) compared to the without-structure condition [31]. Considering the great inter-annual variation of flood occurrence, $B_1(FS_{p_0})$ is typically expressed as the multi-annual mean. Without flood-control structure, a large flood event would result in disruptions of agriculture

and environment, damages to residential properties, public infrastructures and cultural heritages, etc. The degree of downstream damage depends, majorly, on the peak flow and depth of flooding. Denote by $D(p)$ the estimation of flood damage associated with flood of particular occurrence probability $p$. Combining the negative consequences of each potential flood event with the annual probability of such an event, the expected annual damage (EAD) can be calculated adopting the damage-probability curve [32,33], as expressed:

$$EAD = \int_0^1 D(p)\mathrm{d}p \tag{11}$$

where EAD is one of the most frequently used terms to measure flood risk. Note that as downstream flood damage decreases with the increase of flood event probability (i.e., decreasing flood magnitude), there should exist an upper bound of flood event probability (denote by $p_m \in (0,1]$). The expected damage of such magnitude flood ($x_{p_m}$) is negligible and floods larger than this is thus called damaging flood.

The EAD is certainly a theoretical upper limit of flood prevention benefit since a reservoir is impossible to prevent any possible large floods. An attempt to eliminate all flood damage and make downstream flood-free needs so high a cost that few projects are willing to support. The reservoir flood prevention benefit is also intuitively linked to the dam size and the ability to store flood water. For a reservoir withholding flood risk level $p_0$, the flood prevention benefit should be:

$$\mathrm{B}_1\left(FS_{p_0}\right) = \int_{p_0}^{p_m} D(p)\mathrm{d}p. \tag{12}$$

In comparison, the utilization benefit of active storage, denoted by $\mathrm{B}_2\left(AS_{p_0}\right)$, desires to store as much water as possible. For a multipurpose reservoir considering comprehensive utilization, say for hydropower, water supply, recreation, as well as flood control, the active storage benefit is approximately the sum of each possible economic benefit $b_j(\cdot)$, except for the flood control benefit, during the whole flood season.

$$\mathrm{B}_2\left(AS_{p_0}\right) = \sum_{j=1}^{J} b_j\left(AS_{p_0}\right) \tag{13}$$

To find a suitable flood risk level $p_0$, the planner is responsible for maintaining downstream flood safety while caring for the economic benefits arise from active storage. We need to recognize that reservoirs often have multiple objectives, and the simply integrated-welfare maximization may not be applicable for all conditions. Rather, this result is mostly an indication of system design and operation policies. In some cases, political or institutional commitments to a project, if any, are overriding factors. Therefore, a weight factor $\omega$ is introduced. The evaluation of $\omega$ is based on the targets set for large dams, reflecting the commitment bias. The overall objective might be maximizing a weighted combination of the quantifiable annual averaged flood prevention benefit $\mathrm{B}_1\left(FS_{p_0}\right)$ and active storage benefit $\mathrm{B}_2\left(AS_{p_0}\right)$, that is:

$$\max z = \omega \mathrm{B}_1\left(FS_{p_0}\right) + (1 - \omega)\mathrm{B}_2\left(AS_{p_0}\right)$$

s.t.

$$AS_{p_0} + FS_{p_0} = K$$
$$AS_{p_0}, \ FS_{p_0} \geq 0$$

$$\tag{14}$$

The maximum potential values of *AS* and *FS* are obtained in two extreme cases: (1) $AS = K$. The reservoir is not designed to control floods, or reservoir storage capacity is too small, for example, run-of-river dams, to benefit downstream flood safety from the use of flood control storage; (2) $FS = K$. The purpose of flood protection far prevails over other considerations (reservoir is built primarily for flood control purpose), for example, reservoirs located upstream from cities of particularly importance.

### 2.3. Hedging Theory for Refill Period Division

Generally, inflow is higher than the water required for supply or hydropower production in flood season but is much lower in non-flood season. There is a need to restore water in the reservoir at the final stage of flood season thus providing adequate water for non-flood season use. However, whether the reservoir can be fully refilled or not depends largely on inflow condition of the particular year. Specifically, in dry years, the reservoir may be unable to be refilled; while in wet years, the reservoir may be fully refilled in advance and even spilled a lot. From the flood-control point of view, it is the safest to reserve flood control storage from the first flood to the last, however, not economical considering the utilization benefit of reservoir. In view of decreasing inflow and growing water demand after flood season, it is reasonable to assume the date, when the historical average inflow is no longer enough to meet the demand, as the end of refill period (denoted by $t_{r_e}$). However, the problem of when to relax flood-control storage, allowing reservoir to refill, is a problem of hedging. If refilled too early, downstream may face large flood threat; if too late, there may not be enough inflow left for refill. Therefore, the refill circle division is to find a balance between the water conservation benefit at the end of refill period and the increased flood risk during this period.

Denote by $t_{r_s}$ the starting date of refill period. In order to guarantee the high probability of fully refill, the refill period (i.e., $t_r = \int_{t_{r_s}}^{t_{r_e}} dt$) often accepts a certain risk of flood. It can simply suppose that the increased flood risk occurs only when the water level is higher than FLWL. From the damage-probability-curve shown in Figure 2, the occurrence of damaging flood within the refill period should be:

$$P(t_r) = \mathrm{Prob}\left(X \geq x_{p_m}\right) \tag{15}$$

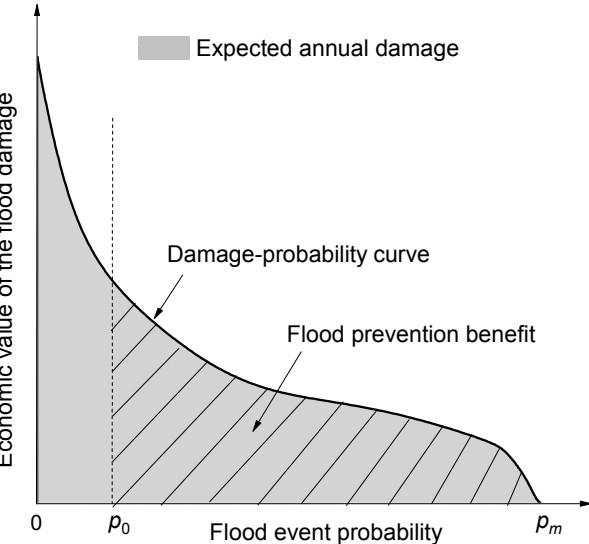

**Figure 2.** Damage-probability curve and expected annual damage.

The loss of flood control benefit due to the increased flood risk thereby can be expressed by the product of the damage probability and the prescribed flood prevention benefit, i.e.,

$$L(t_r) = P(t_r)B_1\left(FS_{p_0}\right) \tag{16}$$

The refill benefit is not confined to the immediate economic benefit but the long-term aftereffects, which rely on the final refilled storage $S_{t_{r_e}}$ at the end of refill period, i.e.,

$$B(t_r) = \sum_{j=1}^{J} b_j\left(S_{t_{r_e}}\right) \propto S_{t_{r_e}} \tag{17}$$

Skipping the main flood season when catastrophic floods occur with high probability, the onset of refill period $t_{r_s}$ should be later than the end of main flood season but earlier than the statistical last flood. This leads to the following mathematical optimization formulation, which maximizes the weighted net water refill benefit and flood control loss in the refill period.

$$\max z = (1 - \omega)B(t_r) - \omega L(t_r)$$

s.t.

$$t_{end\_main\_flood\_season} \leq t_{r_s} < t_{statistical\_last\_flood}$$

$$(18)$$

Equation (18) shows that the potential starting date of refill period $t_{r_s}$ is significantly constrained by the temporal distribution of floods. For river basins where floods have strong regularity, the flood magnitude values in the late flood season could be much smaller than those in the main flood season. Therefore, smaller flood control storages are needed in the late flood season. Under this circumstance, reservoirs can employ the multiple-stage-FLWL instead of annual-FLWL for the whole flood season. This is another means of operation for the sake of refill.

*2.4. Hedging Theory for Drawdown Period Division*

The reservoir drawdown operation differs from the refill operation in that it is devoted to downstream flood safety. The primary restriction factor in drawdown operation is the possible benefit out of the spilled water within the drawdown period. In wet years, the reservoir can maintain high levels throughout non-flood season, and a large amount of water may be spilled when vacating storage to FLWL for early floods in drawdown period, thus resulting in low water utilization benefits. However, in normal or dry years, reservoir water levels before the drawdown period should already be low enough, or even to the dead level, thus almost no economic loss is declared in achieving drawdown operation goal. Usually, the end of drawdown period $t_{d_e}$ should be no later than the beginning of the main flood season, yet the determination of the starting date of drawdown period requires careful consideration. Drawdown too early will lead to a larger probability of spill although higher flood safety. Once the flood season is delayed, i.e., usually started after 1 June but delayed to, say, for example, 20 June, the reservoir may have to operate at low water level. The low water head may cause low efficiency of hydropower generation. Meanwhile, the low rainfall during 1 June to 20 May cause a water shortage. Therefore, the selection of the starting date should be the key to drawdown circle.

Denoted by $t_{d_s}$ the starting date of drawdown period. Similar to the refill period, the flood control benefit loss in the drawdown period (i.e., $t_d = \int_{t_{d_s}}^{t_{d_e}} dt$) is:

$$L_1(t_d) = P(t_d)B_1\left(FS_{p_0}\right)$$

$$(19)$$

The benefit loss (denoted by $L_2(t_d)$) due to the additional water spill can be obtained through simulation compared with the scenario without of drawdown transition period (i.e., $t_{d_s} = t_{d_e}$). Similar to the refill period division, the drawdown period usually begins between the statistical first damaging flood and the start of main flood season. The optimization formulation might be minimizing the weighted sum of flood control benefit loss and spill loss.

$$\min z = (1 - \omega)L_1(t_d) + \omega L_2(t_d)$$

s.t.

$$t_{statistical\_first\_flood} \leq t_{d_s} < t_{start\_main\_flood\_season}$$

$$(20)$$

**3. Danjiangkou Reservoir Case Study**

In this section, a real-world reservoir is employed as a case to demonstrate the connections between hedging theory and the conventional rule curves. The Danjiangkou (DJK) Reservoir (32°37′–32°50′N;

110°25′–111°42′E) is a multi-purpose reservoir in Hubei province, China. It is located in the Han River, the largest tributary of the Yangtze River (see Figure 3).

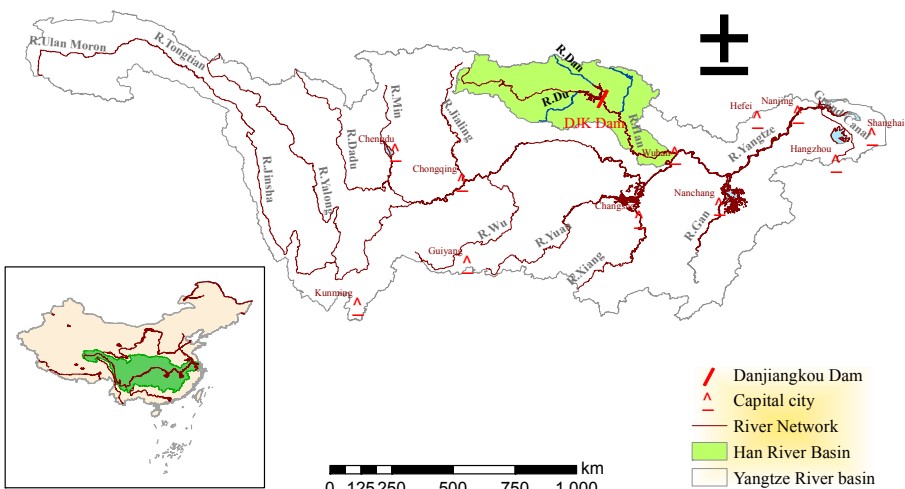

**Figure 3.** Locations of Danjiangkou (DJK) Reservoir and Han River Basin in China.

The DJK Reservoir is the water source of the middle route of the South-to-North water division project (SNWDP) in China. The reservoir drains an area of 95,217 km$^2$ and receives an average annual inflow of 38,780 × 10$^6$ m$^3$, 76.5% of which occurs during May to October. The designed rule curves of DJK Reservoir are shown in Figure 4. The reservoir storages corresponding to dead level (145 m) and normal pool level (170 m) are 10,000 × 10$^6$ m$^3$ and 29,050 × 10$^6$ m$^3$. The operational objectives include water supply, flood control, hydropower generation, and navigation. The primary objective is to secure water supply in the middle route of the SNWDP and downstream water demand. In flood season, the DJK Reservoir takes the task of ensuring the flood safety of the middle and lower reaches of the Yangtze River Basin.

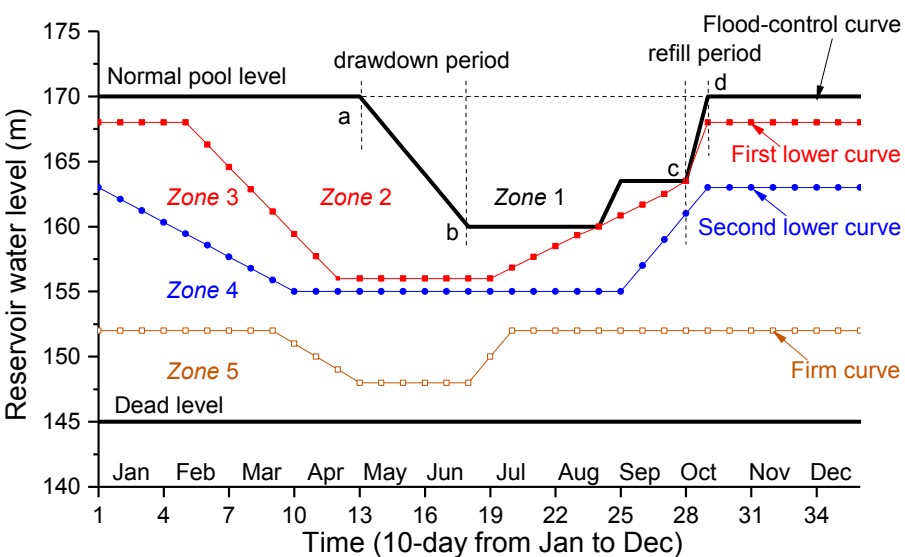

**Figure 4.** The designed rule curves of Danjiangkou Reservoir.

### 3.1. Hedging-Based Water Supply Rule Curves

The water demands assigned to the DJK Reservoir can be split into two types according to water supply priority: Downstream needs mainly for ecosystem (denote as $D_{d,t}$ in the period $t$, about 60% of total needs) and SNWDP transfer (denote as $D_{s,t}$, about 40% of total needs). Downstream needs are

fully satisfied when water level is above 150 m; otherwise cut back by 20%. The operation policy of the SNWDP transfer consists of four curves: Flood control curve, first and second lower supply curve, and firm curve (see Figure 4). Since the transferable discharge in Zone 2 and 3 are close, we combine the two lower supply curves as a target curve for simplicity. The simplified water transfer policy is listed in Table 2; in which, the flood control curve is obtained based on the flood control and refill consideration, we will leave it in the following sub-sections.

**Table 2.** Simplified operating policy of Danjiangkou reservoir for South-to-North water division project (SNWDP) transfer.

| Reservoir Water Level | | Transferable Discharge ($D_{s,t}$, m$^3$/s) |
|---|---|---|
| Above flood-control curve | Zone 1 | 420 |
| Between flood-control and target curves | Zone 2, 3 | 350 |
| Between target and firm curves | Zone 4 | 260 |
| Below firm curve | Zone 5 | 135 |

Following the derivations in Section 2.1, the target and firm curves can be obtained using the zone-based hedging rule curves incorporating the multi-period hedging model. The target of SNWDP transfer is to provide water for the northern provinces of Henan, Hebei, Tianjin, and Beijing. When the transferable water is limited, this amount of water is used mainly to meet the residential living demand such as drinking. Once there is more water being transferred, the extra water can be used for crucial industries or even ecological restoration such as groundwater recharging. Since the water supply for downstream needs ($R_{d,t}$) takes priority over that for SNWDP transfer ($R_{s,t}$), we can treat it as a hard constraint. The water supply reliability of the annually 9.5 billion m$^3$ ($W_{target}$) SNWDP transfer is 95%. Therefore, the water supply optimization target is to maximize the economic SNWDP water supply benefit while meeting the constraints, that is:

$$\text{Max } z = \sum_{t=1}^{T \times N} B(R_{s,t})$$

s.t.

$$S_{t+1} = S_t + I_t - R_{s,t} - R_{d,t} - E_t$$
$$R_{s,t} \leq 420$$
$$\left| \frac{\sum_{t=1}^{T \times N} R_{s,t}/N - W_{target}}{W_{target}} \right| < 5\%$$
$$R_{d,t} = D_{d,t} \quad \text{if } S_t \geq S(150\,\text{m})$$
$$R_{d,t} = 0.8 D_{d,t} \quad \text{if } S_t < S(150\,\text{m})$$

(21)

where $T$ and $N$ are the number of periods ($T$ = 36 10-day) and years ($N$ = 40 years). Constraints considered include continuity equation, South-to-North water division project transferable capacity (420.0 m$^3$/s), downstream water supply hard constraint, and effective reservoir water level boundaries. Given that Beijing is the most important and the last water-recipient city, the unit benefit of transferable water in Zone 5 is represented by the residential water price in Beijing (3.64 Yuan/m$^3$). Likewise, the water price in the nearest SNWDP water supply area, Nanyang section of Henan Province, is 0.18 Yuan/m$^3$, which can be regarded as the unit water benefit in Zone 1. Note that the water supply benefit should be a concave function. Multiplying a discount coefficient, the unit benefits of water in Zone 4 and Zone 2/3 are assumed as 1.82 Yuan/m$^3$ and 0.55 Yuan/m$^3$. Most heuristic optimizations can be easily integrated with simulation models [34–36]. In this paper, the hedging rule curves are obtained by coupling the particle swarm optimization algorithm and an operating simulation model with 40 years (1969–2008) 10-day inflow and demand sequences.

Figure 5 shows the obtained zone-based hedging rule curves and the designed rule curves. The obtained target curve is basically fluctuating around the designed target curve. The obtained firm curve is lower than the one designed in the first half year but approaching in the second half year.

The discrepancy in non-flood season is mainly because the designed firm curve is more conservative. Once the reservoir storage is relatively low, the water supply is artificially reduced in the purpose of avoiding shortage of water transfer. In general, both the target and firm curves are low in spring, but a bit higher in autumn for reservoir refill. The regularity reveals that the conventional rule curves shall be a specific expression of hedging theory with compromising on the engineering risk restrictions. This suggests the consistency between the two.

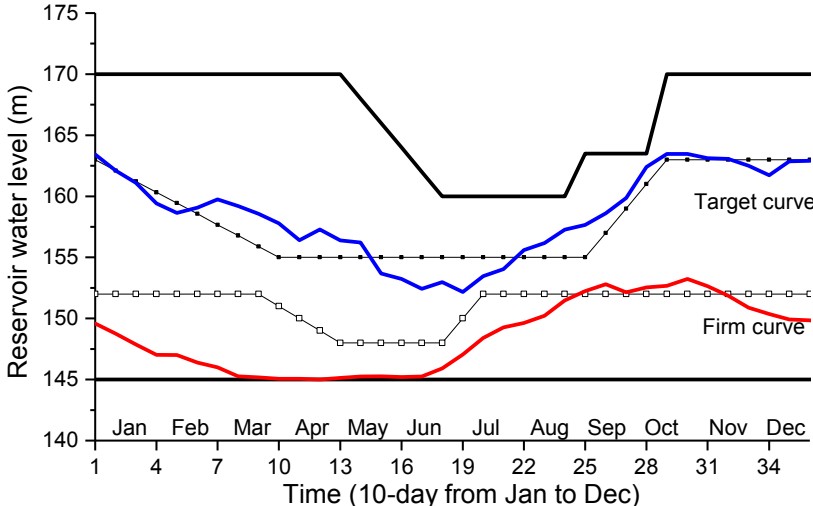

**Figure 5.** The comparison between the best set of zone-based hedging rule curves, in which the blue and red lines are the three-month moving-average target curve and firm curve, and the designed rule curves.

### 3.2. Hedging-Based Flood Limited Water Level

The DJK Reservoir is located in the southeast monsoon zone where precipitation is heavy in summer and autumn. As shown in Figure 4, the entire flood season of DJK Reservoir is divided into main flood season (from 21 June to 20 August) and autumn flood season (from 1 September to 30 September, including the transition period from 21 to 31 August). During the main and autumn flood seasons, reservoir water levels are generally not allowed to exceed the corresponding FLWLs, 160 m and 163.5 m, respectively. This section regards the FLWL in the main flood season (i.e., 160 m from 21 June to 20 August) as an example of how it is determined using hedging theory as described in Section 2.2.

The regulation of DJK Reservoir can significantly reduce the severity of the adverse flood consequences downstream. The procedure for estimating the flood prevention benefit includes the steps: (1) Flood frequency analysis based on the extracted historical floods; (2) estimate the economic loss caused by floods through historical data collection; (3) graph the flood damage-probability curve; and (4) build up the relationship between flood risk level and reservoir flood control storage following the flood routing process.

The flood economic losses estimation usually needs extensive surveys, modeling, and analyses. In this paper, we simply reviewed and summarized the literature (in Chinese) on the historical flooding damages reduction due to the regulation of DJK Reservoir from year 1968 to 1980 [37], year 1983 [38], and year 2010 [39], translating these values into a comparative price level (year 2015) considering the rising of price (i.e., inflation) and economic growth (i.e., GDP). Combined with the flood frequency of the corresponding years, the damage-probability curve is plotted in Figure 6. The expected flood prevention benefit for reservoir of flood safety level $p$ is:

$$B_1\left(FS_p\right) = -\frac{21.03}{5.2}\left[\exp(-5.2) - \exp(-5.2p)\right] \qquad (22)$$

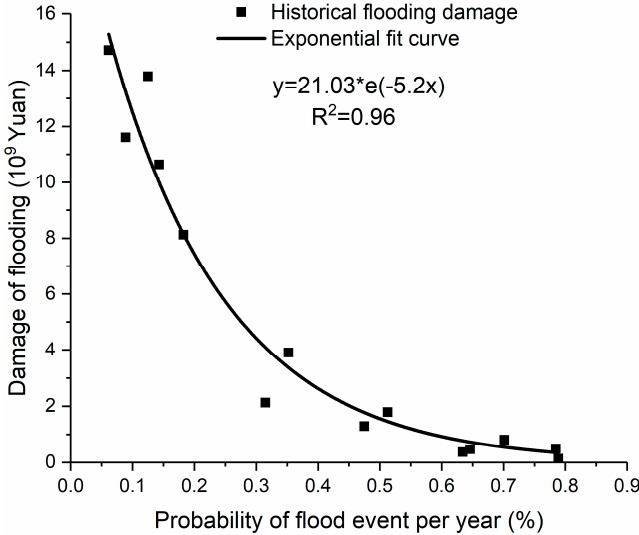

**Figure 6.** Estimated damage-probability curve for the Danjiangkou Reservoir.

Five sets of FLWLs at an increment of 2.0 m are empirically selected and compared. Then it is easy to reversely derive the corresponding risk level and the flood prevention benefit using Equation (22). The active storage benefit is then calculated based on reservoir long-term operation simulation. According to the on-grid electricity price and local water price of Hubei province, set the unit benefit of generated hydropower as 0.3 Yuan/kWh, and water supply benefit as 0.4 Yuan/m$^3$. The unit benefit of water transfer remains the same as that mentioned in Section 3.1. Using the daily inflow and demand in the main flood season from year 1969 to 2008 as inputs, start operation from the selected operating water level. The annual averaged active storage benefit can then be obtained.

The relationship between FLWL scenarios and equal-weighted benefits ($\omega$ = 0.5) is plotted in Figure 7. It can be observed that the total benefit increases with FLWL if it is lower than 160 m but falls if higher, suggesting that the optimal FLWL chosen in the main flood season should be 160 m. Note that the chosen-FLWL scheme may change with the value of weight factor. That is, the larger the weight of flood prevention benefit, the lower the chosen-FLWL. Similarly, the FLWL in the autumn flood season (163.5 m) can also be obtained (not shown because of data limitation).

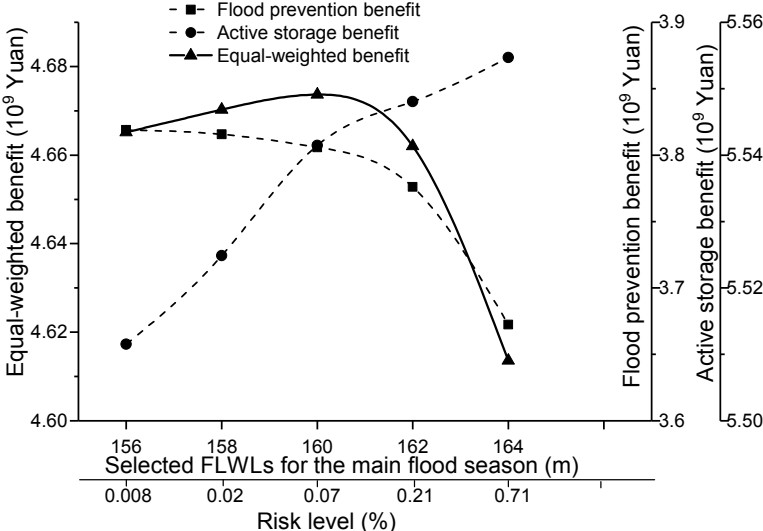

**Figure 7.** Relationship between flood limited water level (FLWL) scenarios, and the results of flood-prevention benefit, mean annual active storage benefit, and the equal-weighted total benefit within the main flood season.

### 3.3. Hedging-Based Refill and Drawdown Periods Design

The refill operation is guided by the upper limited water level. As shown in Figure 4, the DJK Reservoir begins to refill from 163.5 m to 170 m on 1 October, and then keeps the level as high as possible from 10 October to the next April. In order to provide adequate water for non-flood season, a certain risk of flood is acceptable in refill period. Skipping the main flood season (21 June to 20 August) when the large flood occurs with high probability, six refill period schemes are compared. They start on 21 August, 1 September, 11 September, 21 September, 1 October, and 11 October, and all end on 31 December.

The six refill schemes are calculated with the historical 40 years' daily inflow and demand series in the period from 21 August to 31 December. The mean annual refill water levels (refill storage) are then derived. In order to translate these water level into refill benefit, we assume all water above dead level are released for non-flood season use with 60% for downstream needs and 40% for SNWDP, and simultaneously for hydropower generation. For computational convenience and ease of understanding, a linear refill benefit function is employed following Wan et al. [21]. Let the estimated economic value of unit refilled water $u = 1.16$ Yuan/m$^3$. The refill benefit can be calculated, as shown in Figure 8a. From the damage-probability curve in Figure 6, the flood with probability larger than 80% would bring little damage to downstream. Hence, damaging floods should be the flood with the peak flow larger than 8450.0 m$^3$/s. Extract historical damaging floods and then calculate their occurrence probabilities within different refill period schemes. Multiplying these probabilities with the expected flood prevention benefit obtained in Section 3.2, we can get the loss of refill (see Figure 8a). The result shows that the best time of refill operation for DJK Reservoir is 11 September and also vary with the weight factor. This hedging-based refill date is 20 days ahead of the designed date (i.e., 1 October), but very close to the recent researches studying the probable refill rule for DJK Reservoir [40,41]. The discrepancy is mainly because that the reservoir dam has been heightened since 2005 for SNWDP, yet the refill period starting from 1 October is determined on the original storage condition. Currently, the reservoir normal pool level has been raised from 157 m to 170 m and the corresponding storage capacity is enlarged by 66.5%. Under the current reservoir characteristics, an earlier refill is necessary in order to keep the refill efficiency [41].

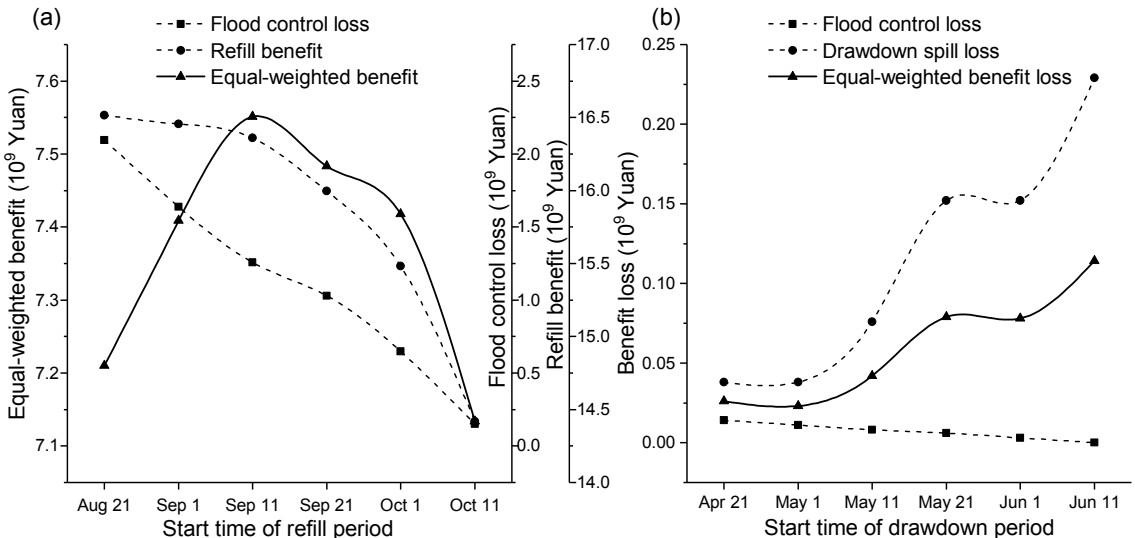

**Figure 8.** Schemes of (**a**) refill starting date and the economic results; (**b**) drawdown starting date and the economic results.

The procedure to obtain the proper time to start drawdown operation is very similar to the refill problem except that the flood control objective takes over the water utilization benefit. The statistic shows that the earliest date that damaging flood might occur is 24 April. Therefore, avoid the main flood season, we schedule six drawdown period schemes that start on 21 April to 11 June using a 10-day interval,

and end on 20 June. The historical daily inflow and demand series in the period from 21 April to 20 June of year 1969 to 2008 are employed to compare the performance of different drawdown schemes. Since the drawdown operation takes effect only in high flow years, it is reasonable to assume that the annual initial regulating water level on 21 April is at the highest water level 170 m. Then the losses of flood control and early-drawdown-induced extra water spill are calculated, as listed in Figure 8b. The result shows that the best time of starting drawdown operation is 1 May, which leads to the minimum benefit loss.

## 4. Rule-Curve-Based Adaptation Strategy Under Non-Stationarity

Both climate change and human activities may aggregate hydrological non-stationarity [9,42]. The reservoir storage or outside characteristic may change due to sedimentation, population and economic growth, groundwater access, new hydraulic structure construction, climate-induced changes on catchment hydrology, etc. For the Han River Basin, Wang et al. [43] used the linear regression and Mann–Kendall method to analyze the long-term trend of the DJK Reservoir inflow. The results show that the annual inflow has a significant downward trend ($p < 0.1$) in the period from 1956 to 2013. Besides, the season inflow and extreme floods have also changed a lot in the past half century [44,45]. Under the growing non-stationary circumstance, rule curves of DJK Reservoir need to be updated and adapt to the changing environment. Section 3 has shown that, in the historical context, conventional rule curves can be seen as an application of hedging theory. Considering that the economic mechanism of hedging theory is comparative clearer whereas the application scope of rule curves is wider, it is reasonable to adjust rule curves for climate change using hedging theory. This section, exemplified by the DJK Reservoir case, identifies the rule-curve-based adaptation strategies under two non-stationary scenarios.

### 4.1. Rule Curves Response to Average Inflow Changes

The scenarios of average inflow changes can generally be divided into two categories, as follows:

*Scenario 1:* More winter snow changes into rainfall as with global warming. Hence, winter runoff increases while spring snowmelt runoff decreases [44]; but the annual average inflow into the DJK Reservoir is unaffected.

Response: Streams and rivers receive water from surface runoff. The changes in surface runoff are logically equivalent to reservoir inflow shifts. As with the unchanged annual average inflow, reservoir water supply reliability and efficiency should stay unchanged basically. The optimal principle is still to deliver identical quantity of water as possible for each period. In spring, if operating following the original rule curves, reservoir water level will drop faster than ever due to the decreased inflow. Once reservoir storage is dropped across water supply zone, operators must reduce water supply according to the zone-based hedging rule curves. This contradicts the optimal principle of hedging rule. To avoid this occurrence, rule curves need to adaptively move downward in spring. Conversely, rule curves need to move upward in winter.

*Scenario 2:* There is a declining trend of annual average inflow [43], but the intra-year inflow distribution is essentially unchanged.

Response: The seasonal fluctuations of zone-based hedging rule curves should remain unaffected as with the unchanged intra-year inflow distribution. There are two adaptive directions to cope with the reduced annual average runoff (i.e., inflow decrease): (1) If the incoming inflow is sufficient to meet the local and SNWDP transfer demands, the optimal principle of equalizing the water supply benefit for each period still holds. The decreased inflow is likely to cause a decrease in reservoir water level. To ensure the principle of equalizing the supply benefit, the zone-based hedging rule curves need to move downward (i.e., rule curves need to adaptively move down). (2) If average inflow is less than water demand, it is impossible to fulfill the water requirements through just rule curves adaptation strategy. The best strategy is to optimize the water use efficiency and promote water-conservation techniques from the demand sides, thus reducing the water supply burden on the reservoir.

The above two scenarios are ideal changing conditions. In fact, studies have found that climate change alters both the intra- and inter- annual inflow of DJK Reservoir. In order to verify the abovementioned adaptive response, a set of future inflow change condition is extracted from Yang et al. [46]. The average 10-day intra-year inflow of the Han River Basin from year 2020 to 2040 under RCP4.5 is shown in Figure 9. Compared with the historical observations, the average annual inflow is reduced by 8.5%, and the seasonal inflow is increased during high flow periods but decreased in low flow periods.

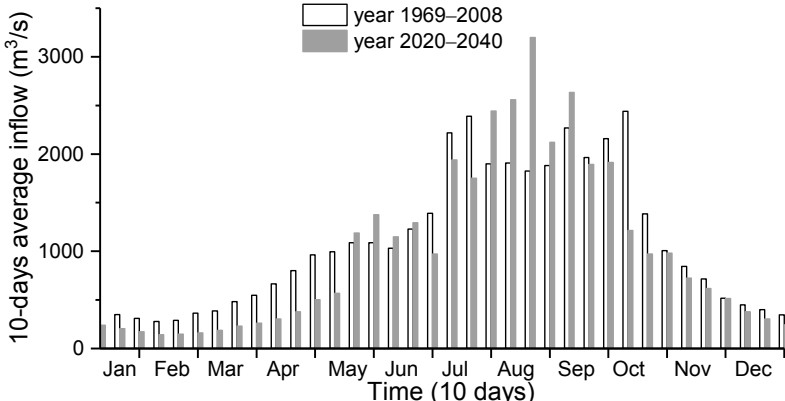

**Figure 9.** Comparison of the historical and predicted future 10-day average inflows into the Danjiangkou Reservoir.

Using this future inflow series as input, the new zone-based hedging rule curves are obtained following the method described in Section 3.1, as shown in Figure 10. The seasonality of the new curves is consistent with that obtained under historical series. On the whole, the target curve moves down by 4.6 m and the firm curve by 2.3 m. The new rule curves move down as with the decrease of annual inflow (as indicated by yellow arrows), which agrees well with the theoretical response in Scenario 2. Comparing the seasonal difference, it is obvious that the shifting degree in spring is larger than the degree in summer (i.e., superposed result of yellow and orange arrows). Specifically, the new target curve between 1 January to 10 May is lower by 6.1 m, but that between 21 July to 10 September is lower by just 2.3 m. The minimum degree in target curve shift occurs in the 22nd 10-day period (i.e.,1 to 10 August). This seasonal difference is in accordance with the response given in Scenario 1.

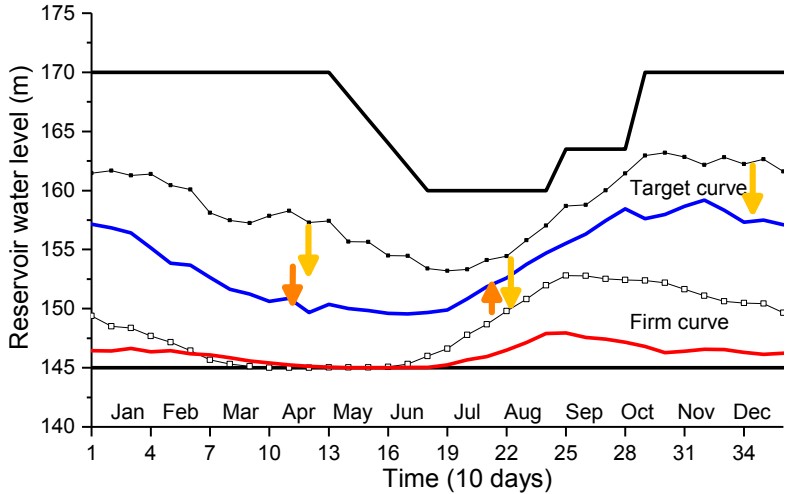

**Figure 10.** The best set of zone-based hedging rule curves under the future scenario (blue and red lines) and the historical series (black lines), where the yellow and orange arrows indicate the qualitative response of rule curves to inflow changes as provided in the responses of Scenarios 1 and 2.

### 4.2. Rule Curves Response to Flood Frequency Changes

The illustrative scenario of flood frequency changes is given as follows:

*Scenario 3:* Owning to the continuous construction of the reservoirs in the upper reaches of the Han River Basin, the flood into DJK Reservoir has been significantly regulated and flattened by upstream reservoirs [45]. The flood frequency in flood season is reduced.

Response: The change of flood frequency mainly affects the flood control problem in flood season. According to the hedging theory for flood control storage allocation, the flood frequency directly determines the flood damage-probability curve: A reduced flood magnitude will lead to a flatter flood peak frequency curve. At the same flood frequency level, the flood magnitude becomes smaller, as does the flood damage. The historical large floods, as well as the corresponding potential damages, will occur with a much lower probability (i.e., lower frequency), resulting in a sharper damage-probability curve. The original flood control storage seems can withstand higher flood magnitude, but actually the flood prevention benefit according to the damage-probability curve is considerably reduced. This effect is roughly equivalent to the effect of reducing the weight coefficient of the flood prevention benefit but increasing the other in Equation (14). Under this condition, the flood control storage should be reduced, i.e., higher FLWL.

Although some studies argue that traditional flood frequency analysis is not applicable under non-stationary environment [47], frequency analysis results are still important references for reservoir design. Yang et al. [48] obtained the flood frequency characteristics under different future climate scenarios for the Han River Basin. They found that the estimated floods at the design frequency level in future climate scenarios are much smaller than those in the historical estimation. The peak flow value of 1000-year return-period flood is reduced from originally 79,000.0 m$^3$/s to 35,550.0 m$^3$/s (in A1B scenario). But for the one-year return-period flood, peak flow value will increase. The flood peak frequency curve is shown in Figure 11a. The rule curve adaptive response example is given on this flood frequency change scenario.

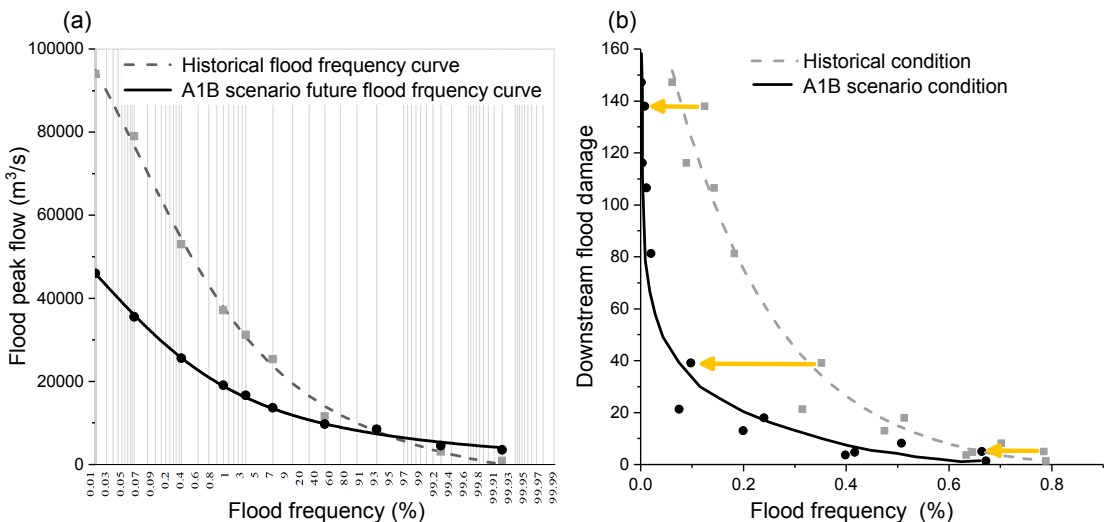

**Figure 11.** Comparison of (**a**) flood peak frequency curves and (**b**) damage-probability curve of Danjiangkou Reservoir, where the yellow arrows indicate the flood frequency qualitative response of historical flood damages to flood frequency changes as provided in the response of Scenario 3.

Based on the established flood-frequency–peak-flow relationship, the flood damage-probability curve is redrawn, as shown in Figure 11b. The new curve is much sharper and significantly shifted to the left. This shift, as indicated by the yellow arrows, is consistent with our adaptive response given in Scenario 3. Under the historical flood condition, the DJK Reservoir is able to regulate flood with peak flow smaller than 79,000.0 m$^3$/s (i.e., flood frequency = 0.7%). However, in the A1B future flood scenario, such magnitude flood occurs over 100 million years, which is mostly not about to happen.

Using the scheme analysis method described in Section 3.2, it is observed that the original flood control storage will provide flood prevention benefit up to the theoretical upper bound of 1.442 billion yuan. With the reduction of flood control storage, the active storage benefits, as well as flood risk, increase. According to the objective that maximizing the weighted benefit of flood prevention and active storage usage, the chosen-FLWL under the new flood condition should be 166 m, which is 6.0 m higher than the original design of 160 m.

## 5. Conclusions

Climate change and human activities aggregate the hydrological non-stationarity, which means that the conventional rule curves need to be updated for adaptation. This paper revisits the conventional rule curves with the risk-based hedging theory from three aspects, i.e., the water supply rule curves pattern, flood-control storage determination, and refill and drawdown periods division. The consistency of rule curves and hedging theory in these aspects is interpreted by mathematical derivation and verified by the Danjiangkou Reservoir example. The rule curves adaptation strategies under two illustrative scenarios are proposed using hedging theory. In this way, one can better understand the proper response of rule curves under different non-stationary scenarios, as well as the rationale of these responses.

Though the rule curves are conventionally designed based on statistical regression and engineering standard with historical information, this paper confirms that behind them there lies the hedging theory. The reservoir hedging theory can be both simple and complex. In essence, hedging is also an optimization method with consideration of uncertainties. At present, most hedging-based reservoir researches constrained uncertainties to hydrological forecast uncertainty [15,20,21]. Yet uncertainties in reservoir operation also have other sources. The design of conventional rule curves does not explicitly express the distribution of natural inflow regime, the function expression of operation benefits, and losses, which actually also have great uncertainties. The conventional rule curves are designed based on these uncertainties and balancing the expected utilities in different operation periods, therefore can be described by general hedging as long as the inputs are consistent with the conventional ones. One of the advantages of hedging theory is that it has clear economic implications and analytical operation principles for different operation objectives, which makes it possible to adapt to the changing environment effectively [18]. This paper finally provides an analytic framework and a demonstrative case to verify this idea. The identified adaptation strategies are straightforward and can be extended for other reservoirs. However, the purpose of these strategies is not to guide real-time operation or replace the ongoing adaptive operation methods. Rather, the strategies are proposed just from the perspective of planning and used as additional source to improve reservoir adaptive capabilities. In the real-time operation, operators should pay more attention to the rolling-updated forecast information [49,50]. The reservoir release decisions are recommended to be dynamically adjusted according to real-time information as well as adaptive rule curves. Therefore, directions for future research involve in two fields: Quantification of the rule-curve-based climate adaptation, and its usability analysis. The analyses in Section 4 should be repeated in depth using more concrete hydrology alteration scenarios, thus providing a flexible tool and quantitative basis for rule curves adaptation in a changing environment. At the same time, the operation risks of the adaptive rule curves will be explicitly quantified. In order to improve the practical application of such adaptive operation, considerably more work is required to cope with the risk-taking problem and persuade the risk-averse reservoir operators and decision makers.

**Author Contributions:** This research was conceived and designed by W.W. in collaboration with all co-authors. Data were collected and analyzed by W.W. The defects of draft were critiqued by J.Z. and J.W. All authors discussed the results and contributed to the final manuscript.

**Funding:** This research was funded by the Key Technologies Research and Development Program (2017YFC0404403, 2016YFC0401302, 2016YFC0402203) and the National Natural Science Foundation of China (91747208, 5171102044, 51579129).

**Conflicts of Interest:** The authors declare no conflict of interest.

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
