# Peer review of "Revisiting Water Supply Rule Curves with Hedging Theory for Climate Change Adaptation"

_sustainability, doi:10.3390/su11071827_

Round 1
Reviewer 1 Report
Comments
(1) The ultimate purpose of this paper is to figure out the rule-curve-based climate adaptation strategies using hedging theory. In Section 4.1, the authors used the average 10-day intra-year inflow of the Han River Basin from year 2020 to 2040 under RCP4.5. if I understand right, the length of future inflow is relatively short, and there is only one set of future inflow under RCP4.5. Such results are not representative. The length of future water inflow should be extended and the water inflow scenario should be increased, such as under RCP 2.6 and RCP 8.5.
(2) The shortest series length for hydrological frequency analysis is 30 years. Data used in Section 4.2 are not explained in detail. As can be seen from Figure 10, there are few points on the curve, which indicate that the authors used short series of future flood to get the flood peak frequency curve. Results obtained in this way at low frequencies are unreliable.
(3) About the results of Section 3.1, the firm curve is not generally approaching the designed rule curve, the authors should give a reasonable explanation about this.
(4) Eq. (2) and Eq. (19) show the immediate benefit (??) and the flood control benefit loss L1(td), but how to calculate the “carryover storage benefits ?(??+1)” and the “spill loss L2(td)” ? The calculation of these two variables is not given in this paper.
(5) Line 140-141 “Since the single period utility function ?(??) is concave, the economic function of storing water for future use ?(??+1) should be concave also” is confusing, more analysis about this should be included.
(6) Line 356-358 “Multiplying a discount coefficient, the unit benefits of water in Zone 4, Zone 2/3 and Zone 1 are assumed as 1.82 Yuan/m3, 0.55 Yuan/m3, 0.18 Yuan/m3, respectively.” The source and basis of these coefficients should be explained in the paper.
(7) In the Eq. (21) and the Eq. (3), do “Lt” and “Et” have the same physical meaning? If so, the authors should use the same symbol.
(8) Some figures and tables in this paper are not standardized. For example, in Figure 9, the labeling of the the yellow and orange arrows is confusing; in Figure 10(b), the coordinate axis lacks units; in Table 2, the “m3/s” should be “m3/s”. The authors should check these figures and tables again to avoid similar error.
(9) Line 418-425, in the process of explaining Figure 3, the authors use specific time to describe the refill period, but readers cannot correspond directly on the abscissa axis. The same problem appears in the process of explaining Figure 4, Figure 8 and Figure 9. It is suggested that the authors add another abscissa (to show dates within a year) to these figures. It is convenient for readers to get more information.

Author Response
We greatly appreciate the comments from the reviewer. We’ve addressed all comments carefully and provided our point-to-point responses in the attached Word file.

Reviewer 2 Report
1. The manuscript presents revisiting water supply rule curves with hedging theory for climate change adaptation, which is interesting. The subject addressed is within the scope of the journal.
2. However, the manuscript, in its present form, contains several weaknesses. Appropriate revisions to the following points should be undertaken in order to justify recommendation for publication.
3. For readers to quickly catch your contribution, it would be better to highlight major difficulties and challenges, and your original achievements to overcome them, in a clearer way in abstract and introduction.
4. It is shown in the reference list that the authors have several publications in this field. This raises some concerns regarding the potential overlap with their previous works. The authors should explicitly state the novel contribution of this work, the similarities and the differences of this work with their previous publications.
5. It is mentioned in p.1 that the hedging theory is adopted for adaptive operation based on hydrological forecasts. What are other feasible alternatives? What are the advantages of adopting this particular approach over others in this case? How will this affect the results? The authors should provide more details on this.
6. It is mentioned in p.1 that the law and design of water supply rule curves, the determination of flood control storage, and the division of refill and drawdown circle are adopted to establish the linkage of rule curves and hedging theory. What are other feasible alternatives? What are the advantages of adopting these particular aspects over others in this case? How will this affect the results? The authors should provide more details on this.
7. It is mentioned in p.5 that “…That is to say, in the high-flow season period, the water supply rule curves (???????,? and ?????,?) shall be low, and vice versa. This explains the…” More justification should be furnished on this issue.
8. It is mentioned in p.9 that a linear refill benefit function following Wan et al. [20] is adopted for refill benefit. What are other feasible alternatives? What are the advantages of adopting this particular function over others in this case? How will this affect the results? The authors should provide more details on this.
9. It is mentioned in p.10 that Danjiangkou Reservoir is adopted as the case study. What are other feasible alternatives? What are the advantages of adopting this particular case study over others in this case? How will this affect the results? The authors should provide more details on this.
10. It is mentioned in p.11 that four curves are adopted for Danjiangkou Reservoir. What are other feasible alternatives? What are the advantages of adopting these particular curves over others in this case? How will this affect the results? The authors should provide more details on this.
11. It is mentioned in p.12 that coupled Particle Swarm Optimization [34] algorithm and an operating simulation model is adopted to obtain the hedging rule curves. What are other feasible alternatives? What are the advantages of adopting this particular soft computing technique over others in this case? How will this affect the results? The authors should provide more details on this.
12. It is mentioned in p.12 that historical records of 1969 to 2008 are taken. Why are more recent data not included in the study? Is there any difficulty in obtaining more recent data? Are there any changes to situation in recent years? What are its effects on the result?
13. It is mentioned in p.14 that six refill period schemes are adopted for comparison. What are the other feasible alternatives? What are the advantages of adopting these particular schemes over others in this case? How will this affect the results? More details should be furnished.
14. It is mentioned in p.15 that “…This hedging-based refill date is quite different with the designed date (i.e., October 1), but very similar to the recent researches studying the probable refill rules for DJK Reservoir [Wang et al., 2014; Duan et al., 2017]. The discrepancy is mainly because…” More justification should be furnished on this issue.
15. t is mentioned in p.16 that three scenarios are adopted for average inflow changes. What are other feasible alternatives? What are the advantages of adopting these particular scenarios over others in this case? How will this affect the results? The authors should provide more details on this.
16. Some key parameters are not mentioned. The rationale on the choice of the particular set of parameters should be explained with more details. Have the authors experimented with other sets of values? What are the sensitivities of these parameters on the results?
17. Some assumptions are stated in various sections. Justifications should be provided on these assumptions. Evaluation on how they will affect the results should be made.
18. The discussion section in the present form is relatively weak and should be strengthened with more details and justifications.
19. There are some occasional grammatical problems within the text. It may need the attention of someone fluent in English language to enhance the readability.
20. Moreover, the manuscript could be substantially improved by relying and citing more on recent literatures about contemporary real-life case studies of soft computing techniques in reservoir operations and/or hydrologic engineering such as the followings:
l Liu B.X., et al., “Parallel Chance-constrained Dynamic Programming for Cascade Hydropower System Operation,” Energy 165 (Part A): 752-767 2018.
l Yaseen, Z.M., et al., “An enhanced extreme learning machine model for river flow forecasting: state-of-the-art, practical applications in water resource engineering area and future research direction,” Journal of Hydrology 569: 387-408 2019.
l Cheng, C.T., et al., “Parallel Discrete Differential Dynamic Programming for Multireservoir Operation,” Environmental Modelling & Software 57: 152-164 2014.
l Taormina, R., et al., “Neural network river forecasting through baseflow separation and binary-coded swarm optimization”, Journal of Hydrology 529 (3): 1788-1797 2015.
l Wu, C.L., et al., “Rainfall-Runoff Modeling Using Artificial Neural Network Coupled with Singular Spectrum Analysis”, Journal of Hydrology 399 (3-4): 394-409 2011.
l Chau, K.W., et al., “Use of Meta-Heuristic Techniques in Rainfall-Runoff Modelling” Water 9(3): article no. 186, 6p 2017.
21. Some inconsistencies and minor errors that needed attention are:
l Replace “…the historical observes…” with “…the historical observations…” in lines 138-139 of p.4
l Replace “…Apply the…” with “…Applying the…” in line 141 of p.4
l Replace “…should great than…” with “…should be greater than …” in line 174 of p.15
l and many more…
22. In the conclusion section, the limitations of this study, suggested improvements of this work and future directions should be highlighted.
Author Response

(The authors gave the same response as above.)

Reviewer 3 Report
Thank you for the opportunity to review the paper, "Revisiting water supply rule curves with hedging theory for climate change adaptation". The issue discussed by the authors is very valid in the context of environmental transformations, primarily caused by climatic changes and intensive human activity. In my opinion, it is a very well written article with a transparent structure and appropriate methodology. I recognise its potential broader application in many other hydrotechnical objects of the kind.
In my opinion, it should be supplemented with a location map (river, reservoir), more morphometric parameters of the Danjiangkou Reservoir should be added, and the discussion should be expanded. I suggest several items, but there are of course considerably more. I encourage the authors to have a detailed insight into e.g. WoS base.
1. Optimal reservoir rule curves under climatic and land use changes for Lampao Dam using Genetic Algorithm, KSCE Journal of Civil Engineering, 2018
2. Multipurpose rule curves for multipurpose reservoir by conditional genetic algorithm, International Review of Civil Engineering, 2018
3. Effect of hedging-integrated rule curves on the performance of the pong reservoir (India) during scenario-neutral climate change perturbations, Water Resources Management, 2016
Author Response

(The authors gave the same response as above.)

Round 2
Reviewer 2 Report
The revised paper has addressed all my previous comments, and I suggest to ACCEPT the paper as it is now.